# Exponential Enhancement of Dark Matter Freezeout Abundance

Bibhushan Shakya

Deutsches Elektronen-Synchrotron DESY, Notkestr. 85, 22607 Hamburg, Germany
bibhushan.shakya@desy.de

October 5, 2022

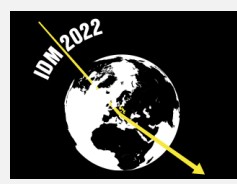

## Abstract

A novel paradigm for thermal dark matter (DM), termed "bouncing dark matter", is presented. In canonical thermal DM scenarios, the DM abundance falls exponentially as the temperature drops below the mass of DM, until thermal freezeout occurs. This note explores a broader class of thermal DM models that are exceptions to this rule, where the DM abundance can deviate from the exponentially falling curve, and even rise exponentially, while in thermal equilibrium. Such scenarios can feature present day DM annihilation cross sections much larger than the canonical thermal target, improving the prospects for indirect detection of DM annihilation signals.

# 1 Introduction

Thermal dark matter (DM) is an attractive and widely studied paradigm, encompassing scenarios where significant interactions keep DM in thermal equilibrium with the thermal bath in the early Universe. In most thermal DM scenarios encountered in textbooks or in the literature (e.g. [1–14]), the DM equilibrium number density at temperatures below its mass is given by the familiar Boltzmann distribution

$$n_i^{\text{eq}} = g_i \left( \frac{m_i T}{2\pi} \right)^{3/2} e^{-m_i/T} \tag{1}$$

where $g_i$ is the number of degrees of freedom of the relevant particle, and $m_i$ is its mass. This distribution is tracked until the rate of interactions that keep DM in equilibrium with the bath become weaker than the Hubble expansion rate of the Universe, known as thermal freezeout, after which the DM number density remains approximately constant. Matching the freezeout abundance with the observed relic density of DM in the Universe yields the standard thermal freezeout cross section $\langle \sigma v \rangle_{\text{canonical}} \approx 2 \times 10^{-26} \text{ cm}^3 \text{ s}^{-1}$ for weak scale dark matter. This is the standard canonical cross section associated with DM with a thermal history, including the popular WIMP paradigm, and serves as a well-defined target for experimental searches of dark matter annihilation.

This exponentially suppressed abundance in Eq. 1, while encountered in most studies of thermal dark matter, is by no means a necessary component of a thermal history. More general classes of thermal DM frameworks can feature stark deviations from this abundance distribution. This broader thermal paradigm has been studied in detail in [15], and termed "bouncing dark matter", referring to behaviour where DM follows the standard distribution Eq. 1 initially but "bounces" to a different (exponentially rising) equilibrium distribution while in thermal equilibrium. Such nontrivial behavior of the number density of a species in thermal equilibrium had previously also been observed in earlier works [16–19], although the associated mechanism was not discussed in detail. This note is only a short summary of the main ideas presented in [15]; the interested reader is referred to that paper for further details.

The bouncing DM framework significantly alters many aspects of early Universe cosmology as well as present day phenomenology of thermal DM. Of these, the most phenomenologically relevant aspect is the enhancement of DM annihilation cross section associated with the correct relic abundance. Recall that in standard thermal histories, the freezeout abundance is inversely correlated with the annihilation cross section; a larger cross section results in DM tracking the equilibrium distirbution Eq. 1 for a longer time, hence freezing out with an abundance that is too small to match the observed DM relic density. However, in bouncing DM frameworks, such larger cross sections can be consistent with the observed abundance thanks to a deviation from Eq. 1 in the later stages of thermal evolution, providing attractive targets far above the canonical "thermal target" for experiments searching for DM annihilation signals.

# 2 Bouncing Dark Matter: Framework

This section discusses the main idea behind the bouncing DM framework. Equilibrium dynamics can be best understood from the chemical potentials of various species in equilibrium, where the chemical potential $\mu_i$ of particle $i$ is defined as $n_i \approx n_i^{\text{eq}} e^{\mu_i/T}$. If an interaction $A+B+... \leftrightarrow Y+Z+...$ is rapid compared to the Hubble parameter, the corresponding chemical potentials are related as

$\mu_A + \mu_B + ... = \mu_Y + \mu_Z + ....$ From this, it becomes immediately obvious that if DM shares the same chemical potential as some lighter species, $A$, in the bath (as occurs in the standard $2 \to 2$ DM freezeout scenarios), it must follow the standard Boltzmann suppressed equilibrium distribution Eq. 1. Therefore, a necessary condition for a deviation from the standard paradigm is that the DM chemical potential must depart from the chemical potentials of all lighter states in the bath. Further, the DM yield $Y_\chi = n_\chi/s$ (where $s = 2\pi^2 g_* T^3/45$ is the total entropy density, and $g_*$ is the effective number of degrees of freedom in the bath) can not only deviate from Eq. 1, but in fact rise, if $\mu_{DM}$ satisfies

$$\mu_\chi(x) + x\frac{d\mu_\chi(x)}{dx} > m_\chi\left(1 - \frac{3}{2x}\right), \tag{2}$$

where $x = m_{DM}/T$. In any configuration that satisfies Eq. 2, the DM number density will deviate from the canonical distribution Eq. 1 and rise instead of following exponentially while in thermal equilibrium.

## 3 Bouncing Dark Matter: Example

This section briefly discusses an example that realise bouncing dark matter. The interested reader is referred to the original reference [15] for further details.

Consider a dark sector with three scalar particles, the DM candidate $\chi$ and two other states $\phi_1, \phi_2$, that interact via

$$-\mathcal{L} \supset \lambda_{\chi 1}\chi^2\phi_1^2 + \lambda_{\chi 2}\chi^2\phi_2^2 + \lambda_{12}\phi_1^2\phi_2^2 + \lambda\phi_2^2\chi\phi_1. \tag{3}$$

Further, assume the masses follow the relations

$$m_\chi > m_{\phi_2} > m_{\phi_1}, \qquad 2m_{\phi_2} > m_\chi + m_{\phi_1}. \tag{4}$$

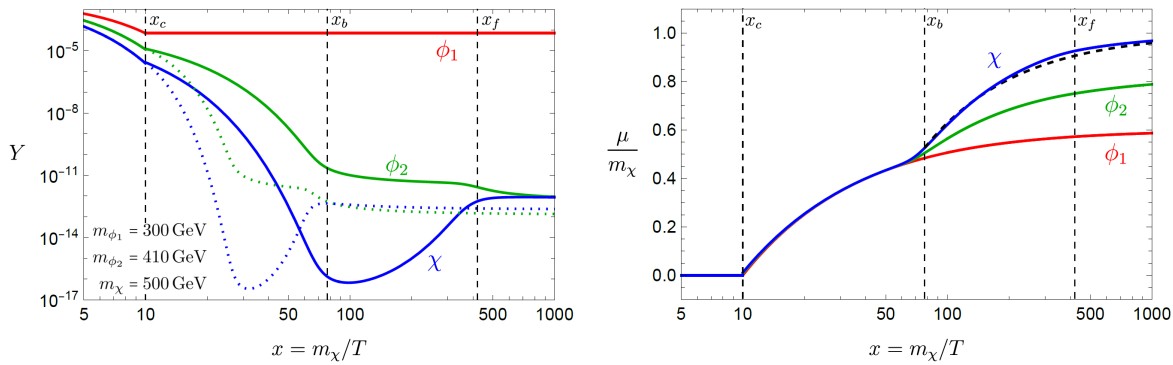

Figure 1: Evolution of yields of dark sector particles (left) and their chemical potentials (right). The bounce, denoted by $x_b$, refers to the point where $\phi_2\phi_2 \to \chi\phi_1$ becomes the dominant process, as all other processes freeze out; this allows the DM chemical potential to deviate from those of the other species (right panel), enabling the DM abundance to deviate from the standard distribution, and rise exponentially. See the original reference [15] for details.

If all interactions are rapid, the chemical potentials are related as $\mu_\chi = \mu_{\phi_1} = \mu_{\phi_2}$, and DM follows the standard distribution. However, if the final stages of freezeout is controlled by the

process $\phi_2 \phi_2 \to \chi \phi_1$, the relation is instead $\mu_\chi + \mu_{\phi_1} = 2\mu_{\phi_2}$, which allows $\mu_\chi$ to deviate from the rest, allowing the DM abundance to depart from the standard Boltzmann distribution even while in equilibrium. This is illustrated for a benchmark point in Fig. 1 (see [15] for details). Note that the second relation in Eq. 4 is needed to ensure DM stability, as well as to make DM production kinetmatically favorable in the later stages of evolution, which drives the bounce.

While this is only one example, there are several other frameworks that can realize the bouncing DM mechanism in qualitatively different ways, such as from the decay of the coannihilating partner, or from $3 \to 2$ processes. Such variations are discussed in detail in [15].

## 4 Bouncing Dark Matter: Phenomenology

In the example discussed above, DM can annihilate in the present Universe as $\chi \chi \to \phi_i \phi_i$, where $\phi_1$ can subsequently decay into Standard Model (SM) final states to yield observable DM signals. Fig. 2 shows the predicted DM annihilation cross sections consistent with the observed relic density for various parameter choices (see caption for details). For comparison, the figure also shows current limits from Fermi data, as well as the predicted reach for the Cherenkov Telescope Array (CTA), assuming $\phi_1 \to WW$, adapted from the results in [20].

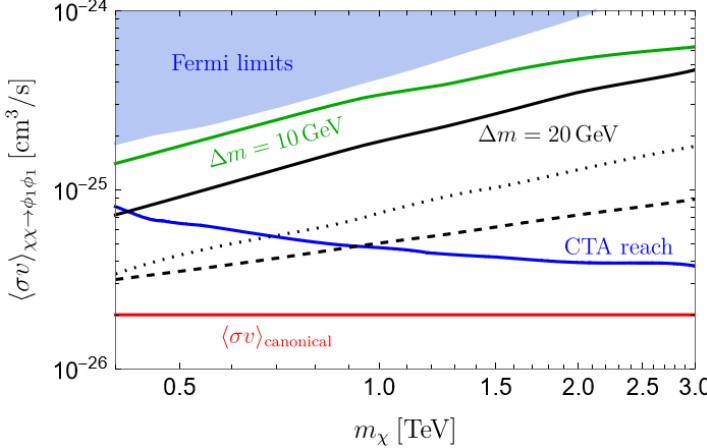

Figure 2: Predicted present day $\chi \chi \to \phi_1 \phi_1$ annihilation cross section (consistent with the observed DM relic density) as a function of $m_\chi$, with $m_{\phi_1} = m_\chi/2$ and $\Delta m = m_{\phi_2} - (m_\chi + m_{\phi_1})/2 = 10, 20 \, \text{GeV}$ (green, black curves respectively). The solid, dotted, and dashed curves correspond to various assumptions about kinetic decoupling between the dark and SM sectors (see [15] for other details of the figure). The blue shaded region and the solid blue curve denote Fermi bounds and CTA projected reach (for $\phi_1 \to WW$). The canonical thermal target $\langle \sigma v \rangle_{\text{canonical}}$ (red line) is also shown for reference.

The figure clearly illustrates that the bouncing DM framework can allow present day DM annihilation cross sections that are much larger than the canonical thermal target. For the benchmark points chosen in the figure, upcoming experiments (CTA) would be unable to probe the canonical thermal target, but can observe signals from bouncing DM, highlighting the improved prospects of DM discovery.

## 5 Conclusion

While a Boltzmann suppressed number density is taken to be a standard component of thermal dark matter scenarios, this notes highlights a more general class of dark matter models, *bouncing dark matter*, that allows for deviations from this standard behavior. Such scenarios are characterized by the presence of various interactions in the final stages of thermal evolution that allow the dark matter chemical potential to deviate from those of the other particles in the bath, thereby triggering a transition to an exponentially rising abundance distribution. Such behaviour results in an exponential enhancement of DM freezeout abundance, and can be readily realized in realistic beyond the Standard model (BSM) setups. Phenomenologically, bouncing DM is an unusual scenario that can consistently accommodate DM annihilation cross sections much larger than the canonical thermal target, improving the prospects for indirect detection of DM signals.

## Acknowledgements

The author is supported by the Deutsche Forschungsgemeinschaft under Germany's Excellence Strategy - EXC 2121 Quantum Universe - 390833306.

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
