# Peer review of "Exponential Enhancement of Dark Matter Freezeout Abundance"

_SciPost Physics Proceedings_

## Round 1 · Referee Report · Anonymous (Referee 1) · 2022-10-21

Strengths
-
The manuscript studies a novel mechanism of dark matter freeze-out, by introducing certain mass hierarchy among several (meta-)stable dark particles.
-
It clearly illustrates the effects of such mass hierarchy in dark matter freeze-out, especially it requires a slightly larger dark matter annihilation cross section than the standard case.
Weaknesses
Report
Requested changes
I strongly suggest that the author mention the effect of other processes, e.g. $\chi + \phi_2 \to \phi_1 + \phi_2$, as this puzzles me a lot during reading this proceeding. It seems to forbid $Y_{\chi} > Y_{\phi_2}$, if I understand their original paper correctly.

Author: Bibhushan Shakya on 2022-12-20 [id 3164]
(in reply to Report 1 on 2022-10-21)Thank you for the report. A revised version addressing the referee's comment will be resubmitted.

---

## Editorial Decision

unknown